# Presence of Antibodies Binding to Negative Elongation Factor E in Sarcoidosis

**DOI:** 10.3390/jcm9030715

**Published:** 2020-03-06

**Authors:** Niklas Baerlecken, Nils Pursche, Torsten Witte, Katja Kniesch, Marius Höpfner, Diana Ernst, Frank Moosig, Benjamin Seeliger, Antje Prasse

**Affiliations:** 1Private Practice Rheumatology, 50996 Cologne, Germany; baerleckennt@gmail.com; 2Department of Clinical Immunology and Rheumatology, Hannover Medical School, 30625 Hannover, Germany; nils.pursche@gmx.de (N.P.); Witte.Torsten@mh-hannover.de (T.W.); Kniesch.Katja@mh-hannover.de (K.K.); Hoepfner.Marius@mh-hannover.de (M.H.); ernst.diana@mh-hannover.de (D.E.); 3Rheumazentrum Schleswig-Holstein Mitte, 24534 Neumünster, Germany; moosig@rheuma-sh.de; 4Department of Respiratory Medicine, Hannover Medical School and Biomedical Research in End-stage and Obstructive Lung Disease Hannover, German Lung Research Center (DZL), 30625 Hannover, Germany; seeliger.benjamin@mh-hannover.de; 5Fraunhofer Institute for Toxicology and Experimental Medicine, 30625 Hannover, Germany

**Keywords:** sarcoidosis, autoantibodies, autoimmunity, granuloma, ELISA

## Abstract

Sarcoidosis is characterized by multiorgan involvement and granulomatous inflammation. Its origin is unknown and the potential role of autoimmunity has not been sufficiently determined. We investigated the presence of autoantibodies in sarcoidosis using protein array technology. The derivation cohort consisted of patients with sarcoidosis (*n* = 25) and controls including autoimmune disease and blood donors (*n* = 246). In addition, we tested a validation cohort including pulmonary sarcoidosis patients (*n* = 58) and healthy controls (*n* = 13). Initially, sera of three patients with sarcoidosis were screened using a protein array with 28.000 proteins against controls. Thereby we identified the Negative Elongation Factor E (NELF-E) as an autoantigen. With confirmatory Enzyme-linked Immunosorbent Assay (ELISA)testing, 29/82 patients (35%) with sarcoidosis had antibodies against NELF-E of the Immunoglobulin (Ig) G type, whereas 18/253 (7%) sera of the controls were positive for NELF-E. Clinically, there was an association of the frequency of NELF-E antibody detection with lung parenchymal involvement and corresponding x-ray types. NELF-E autoantibodies are associated with sarcoidosis and should be further investigated.

## 1. Introduction

Sarcoidosis is a systemic disease of yet undetermined cause and is mainly characterized by multiorgan involvement with granulomatous inflammation [1,2,3]. Both genetic and environmental factors have been identified to play a role in the pathogenesis of its disease [4]. Sarcoidosis can affect all organs, but the main organ involvement are lymph nodes, lung, skin, eyes, and joints [5]. Its origin is unknown. However, certain agents, viruses, and bacteria have been reported that they may trigger sarcoidosis, e.g., pine tree pollen, clay, zirconium, talc, aluminum, herpes virus, Epstein-Barr virus, retrovirus, coxsackie B virus, cytomegalovirus, Borrelia burgdorferi, Propionibacterium acnes, mycobacterium tuberculosis, and atypical mycobacteria [6,7,8,9,10,11,12]. 

Despite numerous attempts, only circumstantial evidence exists to regard sarcoidosis as an autoimmune disease, including its response to immunosuppressive therapy, association with certain Human Leucocyte Antigen (HLA) genotypes and detection of some autoantibodies that are also associated with other autoimmune diseases such as antimitochondrial, antinuclear, anti-double stranded deoxyribonucleic acid (DNA) and citrullinated cyclic peptide antibodies [13].

Recent advances in protein array technology offers the opportunity for unbiased high-throughput screening of autoantibodies in sera. Protein arrays have been used repeatedly for the detection for autoantibodies in autoimmune diseases and malignant diseases [14]. In the past 20 years, planar and bead-based protein-arrays with recombinant or non-recombinant peptides or proteins have been established. They allow a simultaneous screening of 1000-100.000 potential autoantigens. Our used protein-array is a planar protein array using an Escherichia coli host system and containing 28.000 different full and partial recombinant proteins of a fetal human brain complementary DNA. By using this system, we could identify autoantibodies in giant cell arteritis, polymyalgia rheumatica, Takayasu arteritis, spondyloarthritis, arteriosclerosis, complex regional pain syndrome, and primary Sjögren syndrome with polyneuropathy [15,16,17,18,19,20].

With the aid of this established screening platform using protein array technology, we identified autoantibodies in a cohort of patients with different types of autoimmune diseases or sarcoidosis. We found autoantibodies against the ribonucleic acid (RNA) binding protein Negative Elongation Factor E (NELF-E) significantly associated with sarcoidosis and pulmonary involvement. This is the first study demonstrating the presence of NELF-E antibodies in a specific disease. Further studies are needed to clarify the exact role of NELF-E autoantibodies in sarcoidosis.

## 2. Experimental Section

### 2.1. Patient Selection and Clinical Data

For the derivation cohort we collected the sera of patients with sarcoidosis (*n* = 24), patients with systemic lupus erythematosus (SLE, *n* = 36), patients with febrile infectious diseases (FID, *n* = 46), patients who received a positron-emission tomography test (PET controls (PET, *n* = 64), and healthy blood donors (BD, *n* = 100). Patients with sarcoidosis and SLE were recruited from the rheumatological clinic and hospital ward from 2011 to 2013. A second validation cohort of sarcoid patients was recruited from the interstitial lung disease outpatient clinic (*n* = 58) and healthy controls were recruited from our departments (*n* = 13). The study was conducted in accordance with the ethical standards laid down in the 1964 Declaration of Helsinki and its later amendments were approved by our local ethical committee (project number 4928, approval date April 10th 2008), and all patients provided informed consent.

Sarcoidosis was diagnosed by rheumatologists or pulmonologists based on typical clinical and histological findings in accordance with the consensus statement of the American Thoracic Society, European Respiratory Society (ERS) and World Association of Sarcoidosis and Other Granulomatous Disorders and SLE according to the former American College of Rheumatology criteria [21,22]. Chest x-rays of patients with sarcoidosis were evaluated using the Scadding scale [23]. The PET controls consisted of patients with pulmonary diseases excluding sarcoidosis from a previous cohort: 32/64 PET controls had different underlying malignant diseases, 13/64 had lung fibrosis, and 19/64 had chronic obstructive pulmonary disease (COPD) [18].

Clinical data were collected including available immunological data on antinuclear antibodies and extractable nuclear antigens, disease manifestation and course, histology, imaging, and treatment.

### 2.2. Protein Array

Three different sera of patients with acute sarcoidosis presenting with Loefgren’s syndrome at disease onset were screened for the presence of autoantibodies by high-density protein array (human cDNA expression library (hEx1), Source BioScience LifeSciences, Nottingham, Nottinghamshire, United Kingdom) and subsequently compared against existing screens of patients with inflammatory or autoimmune disease. These patients were initially chosen due to their characteristic clinical features and brief disease history. The protein array is based on a cDNA of fetal brain tissue (Lib. No. 800). The clones were expressed by Escherichia coli. The screening was performed according to the protocol of Source BioScience LifeSciences. Sera of the patients were separately added to protein arrays. After overnight incubation, a second alkaline phosphatase labelled antibody goat anti-human Immunoglobuline G (IgG) (Fragment crystallizable-region (Fc) specific, Sigma Aldrich Corporation, St. Louis, MO, USA) in a dilution of 1:5000 was added to the protein array. Using a Typhoon 9400 scanner (GE Healthcare, Chicago, IL, USA) and ECF (GE Healthcare, Chicago, IL, USA) results could be imaged. The pictures were analyzed by Adobe Photoshop Creative Suite 3 (Adobe Systems Incorporated, version 10.0.1, St. Jose, CA, USA). Subsequently, marked dots can be identified via an electronic library provided by imaGenes and compared to previous results in other diseases.

### 2.3. Enzyme-linked Immunosorbent Assay (ELISA) for Detection of IgG Antibodies Binding to Negative Elongation Factor E (NELF-E)

For performing the ELISA tests, 96-well plates (Nunc Maxisorb, Thermofisher, Waltham, MA, USA) were coated with 100 µg of the NELF-E peptide (synthesized by Biomatik, Wilmington, Dalaware, USA) overnight: Asn-Thr-Leu-Tyr-Val-Tyr-Gly-Glu-Asp-Met-Thr-Pro-Thr-Leu-Leu-Arg-Gly-Ala-Phe-Ser-Pro-Phe-Gly-Asn-Ile-Ile-Asp-Leu-Ser-Met. The next day, the coating solution was skipped and the plates were blocked with 300 µL per well 5% bovine albumin serum (BSA) for 1 h. The blocking solution was skipped and the plates were incubated with sera in a 1:100 dilution in 20% BSA phosphate-buffered saline (PBS) for 30 min at room temperature. For the definition of a cut-off, sera of 16 patients with sarcoidosis, 16 with infectious diseases and 16 blood donors were measured in duplicate. The optimal cut-off to differentiate sarcoidosis patients and controls was measured by Receiver-operating characteristics (ROC) analysis. The serum of a sarcoidosis patient which provided an optical density (OD) exactly at that cut-off was used as cut-off control in all further assays. The concentration of antibodies in this serum was defined as 1 (arbitrary unit) for IgG-type antibodies against NELF-E. After 30 min of incubation, the plates were washed 3 times with PBS. Next, 100 µL of a secondary peroxidase-goat anti-human IgG antibody labeled with horseradish peroxidase (HRP) (Jackson ImmunoResearch Europe Ltd., Ely, Cambridgeshire, UK) was added in a dilution of 2:10,000 or 2 µL in 8 mL PBS. The plates were incubated for 30 min at room temperature and washed 3 times with PBS. The color reaction was performed with tetramethylbenzidine (TMB) (Therma Scientific, Waltham, MA, USA) for up to 15 min according to the manufacturer’s instructions and the ODs were read at 450 nm in an ELISA reader.

### 2.4. Statistical Analysis

Wilcoxon- and Kruskal-Wallis tests were used to evaluate differences in serum IgG concentrations and Fisher’s exact or Chi²-test were used to evaluate serum IgG cut-offs within disease groups and by clinical characteristics, as appropriate. All *p* values refer to two-tailed tests. Two-tailed *p* < 0.05 was regarded as statistically significant. Analyses and graphs production were performed using STATA V16 (STATA Corp LP, College Station, TX, USA) RStudio version 1.2.5033 (RStudio Inc, Boston, MA, USA).

## 3. Results

### 3.1. Sarcoidosis Cohort

We included a total of 82 patients with a definite diagnosis of sarcoidosis (first derivation cohort *n* = 24; second validation cohort *n* = 58), 43% of which were female with a median age of 48 years (interquartile range (IQR) 39–59). The majority of 88% had pulmonary involvement (89% with parenchymal involvement of the lung defined by an X-ray stage by Scadding Scale ≥ 2). Eighty-four percent had a chronic or relapsing disease with need for continued immunosuppressive therapy for at least 2 years. The remaining characteristics are displayed in Table 1.

### 3.2. Protein Array Results

The sera of each patient with sarcoidosis bound to a number of autoantigens (Figure 1). One autoantigen was shared by all 3 screening sera of patients with sarcoidosis: Negative elongation factor E (NELF-E; RNA-binding protein RD) (Figure 2). We compared our results with former protein array screenings, which were performed in former studies [13,14,15,16,17,18]. NELF-E antibodies were not detected in any of these protein screening arrays from patients with GPA (*n* = 4), CSS (*n* = 15), RA (*n* = 12), tuberculosis (*n* = 4), AOSD (*n* = 3), B-NHL (*n* = 4), Morbus Ormond/IgG4-related disease (*n* = 4), giant cell arteritis (*n* = 4).

### 3.3. IgG-Antibodies Binding to NELF-E

In the pooled cohort, 29/82 (35%) patients with sarcoidosis had antibodies against NELF-E of the IgG type (10/24 in the initial derivation cohort (42%) and 19/58 in the validation cohort (33%). 3/46 (6.5%) patients with febrile infectious disease (FID), 4/36 (11%) patients with systemic lupus erythematosus (SLE), 3/64 (4.7) positron emission tomography (PET) controls, 8/100 (8%) of blood donors (BD) and 0/13 healthy controls were positive for NELF-E (Figure 3).

### 3.4. Clinical Association of Anti-NELF-E Antibody Detection

Patients that had detectable anti-NELF-E IgG antibodies in the serum showed more involvement of the lung parenchyma compared to negative patients with regards to X-ray types (type 0 or 1 present in 4/17 (24%) anti-NELF-E positive patients vs. 25/64 (39%) (Fisher’s exact *p* = 0.041). The median antibody titer was significantly higher with lung parenchymal involvement (0.98 (IQR 0.89–1.02) vs. 0.77 (IQR 0.44–0.91), *p* = 0.0053) (Figure 4A and 4B). Neither age, extrapulmonary involvement or chronic disease course were different among the groups (Table 1). 

## 4. Discussion

We could show that antibodies binding to NELF-E of the IgG type are present in a subset of patients with sarcoidosis, particularly in those with lung parenchymal involvement, but not in controls including blood donors, systemic lupus erythematosus and other inflammatory/rheumatologic diseases. To our knowledge, the presence of NELF-E antibodies has not been described in the context of other diseases to date.

While we recruited a fairly large number of sarcoidosis patients, only about 35% of them were found to produce anti-NELF-E antibodies. Recently, there were 5 clinical clusters described within the spectrum of sarcoidosis: abdominal; ocular–cardio–cutaneous–central nervous system; musculoskeletal–cutaneous; pulmonary–lymphonodal; extrapulmonary [5]. Given the recruitment of rheumatology and respiratory outpatient units, the abdominal, and ocular-cardio-cutaneous-central nervous system clusters are underrepresented, so the specificity towards certain organ involvement remains to be refined. Additionally, a significant proportion of our patients were not treatment-naïve at the time of serum collection and the effect of immunosuppressive therapy on prolonged autoantibody positivity cannot be reliably assessed in this cohort, although patients with Loefgren’s syndrome were less frequently positive in this cohort. Furthermore, in many cases, tissue samples were not available for review by a reference pathologist and data review was limited to written reports so sound differences between the NELF-E antibody positive and negative cohorts in this regard cannot be reliably assessed.

NELF-E is part of the NELF complex. NELF-E shows the strongest RNA binding activity of the NELF complex and may start recruiting the NELF complex to RNA. NELF, a multisubunit complex containing RD, cooperates with DSIF to repress RNA polymerase II elongation [24,25,26]. Narita et al. described NELF-E expression as widely expressed, especially in heart, brain, lung, placenta, liver, skeletal muscle, kidney, and pancreas [27].

Little is known considering NELF-E in autoimmune diseases or sarcoidosis. Hočevar et al. published recently a statistical analysis of genome-wide association study (GWAS). GWAS signals could be found increased in the region of NELF-E, SKIV2L, and STK19 genes [28]. In our study, we found only for NELF-E but not for SKIV2L or STK19 autoantibodies in sarcoid patients. It is quite rare that genetic differences lead to the development of autoantibodies against the product of the same gene. In the cancer field, however, it is well recognized that autoantibodies against driver mutations/oncogenes precede the manifestation of the cancer and may be used as a predictive biomarker [29]. It is noteworthy that NELF-E is not expressed as an extracellular antigen and thus cannot be targeted by the immune system with the tissue being intact. NELF-E is likely identified as an antigen once exposed within inflamed and necrotic tissue since it is highly expressed NELF-E within the nucleus.

Sarcoidosis is not considered as an autoimmune disease but may have autoimmune features. In some patients, suffering from autoimmune diseases and sarcoidosis presence of autoantibodies have been also reported. Grunewald et al. identified lung-accumulated Vα2.3+Vβ22+ T-cell clones in association with HLA-DRB1*03 molecules, and inter-patient similarities between both T-cell receptor (TCR) α and β chain sequences which might indicate a specific antigen in sarcoidosis [30]. Besides, a vimentin peptide was identified as a T-cell antigen. However, nothing has been reported considering its diagnostic value. Kinloch et al. investigated further vimentin as an antigen, but anti-vimentin antibodies serum titers could not discriminate between sarcoidosis and healthy controls [31]. Moreover, in one study antinuclear antibodies (ANA)-titers were observed in 28.5% of the sarcoid patients but potentially underlying autoantibodies were not further specified [32]. These positive ANA-titers could be partly the result of antibodies directed against NELF-E. Thus, our study is the first describing the high prevalence of NELF-E autoantibodies in sarcoid patients.

It remains unclear whether NELF-E autoantibodies have any functional consequences or are merely a bystander phenomenon. Preliminary, yet unpublished data, suggests that NELF-E autoantibodies may develop because of a mimicry effect between a certain part of the amino sequence of NELF-E and Mycobacteroides abscessus and other bacteria, which would fall in line with previously published associations between sarcoid and these pathogens [12]. Further research is required to establish the precise functional role of NELF-E antibodies in sarcoidosis. We fully acknowledge that our findings at this stage are primarily hypothesis-generating and do not establish a clear link between autoimmunity and sarcoidosis, and given their presence in only a subset of sarcoidosis patients, cannot function as a diagnostic tool at this stage.

## 5. Conclusions

NELF-E autoantibodies are associated with sarcoidosis and should be further investigated.

## Figures and Tables

**Figure 1 jcm-09-00715-f001:**
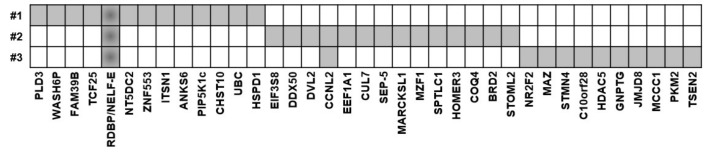
Positive autoantigens in the three screening sera from patients with sarcoidosis.

**Figure 2 jcm-09-00715-f002:**
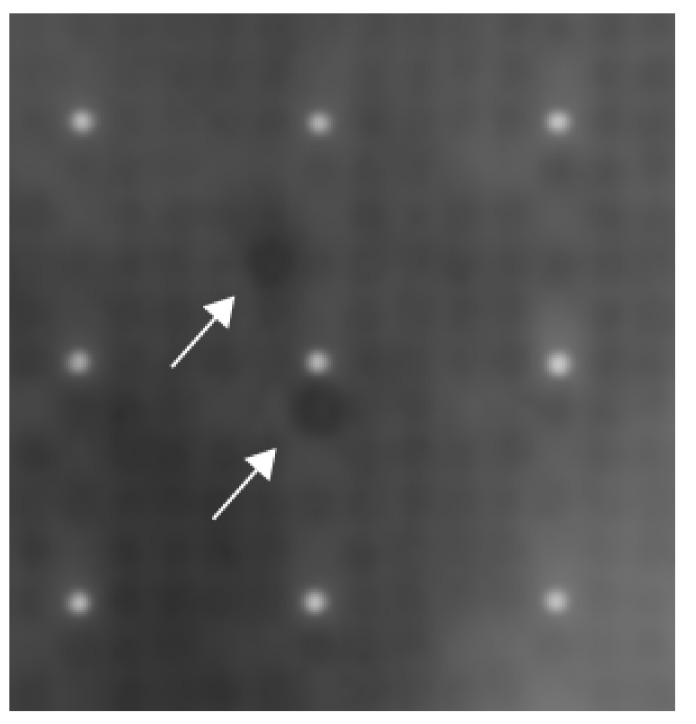
Exemplary result of one of the three protein-arrays using the sera of patients with sarcoidosis. Each white dot is surrounded by 24 grey dots and together they form a cluster within the protein array. The grey dots contain proteins. Each protein can be found at least two times being apart as far as possible in one cluster. In the center of the picture, two dark spots can be spotted. These two dark spots are Negative Elongation Factor E (white arrows).

**Figure 3 jcm-09-00715-f003:**
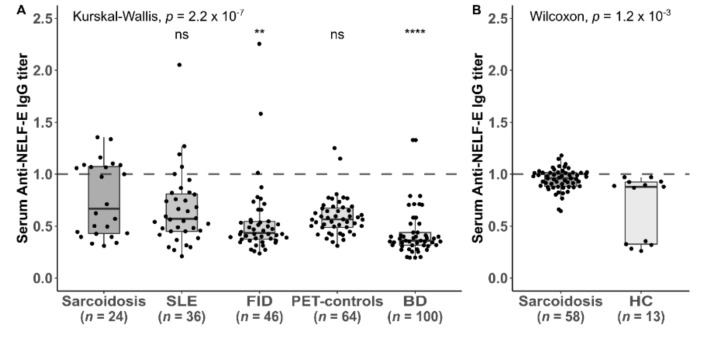
Titers of IgG-antibodies against the Negative Elongation Factor E across different diseases in two different cohorts: (**A**) initial derivation cohort including patients with sarcoidosis, systemic lupus erythematosus (SLE), febrile infectious disease (FID), patients with miscellaneous diseases within the positron emission tomography (PET)-controls and blood donors (BD); (**B**) validation cohort including patients with sarcoidosis (*n* = 58) and healthy controls (HC) (*n* = 13). The titer is considered positive, if it is the same or higher than the cut-off serum (dashed line. Global Kruskal-Wallis is shown, as well as pairwise Wilcoxon-tests of disease groups vs. the sarcoidosis group (**** indicates *p* < 0.0001; ** indicates *p* < 0.01, ns indicates *p* ≥ 0.05).

**Figure 4 jcm-09-00715-f004:**
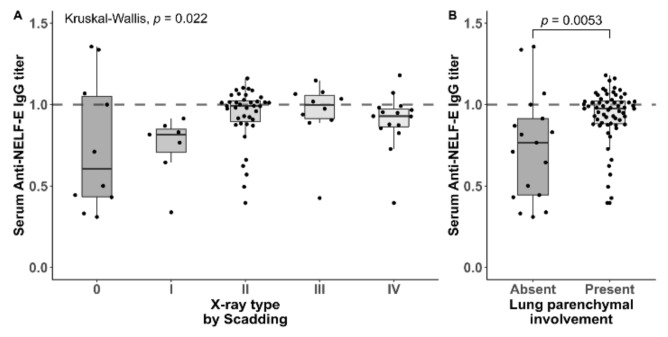
(**A**) Titers of Immunoglobuline G-antibodies against the Negative Elongation Factor E across X-ray types by Scadding scale in patients with sarcoidosis (*n* = 82). (**B**) Titers of IgG-antibodies against the NELF-E depending on lung parenchymal involvement in patients with sarcoidosis (*n* = 82). The titer is considered positive, if it is the same or higher than the cut-off serum.

**Table 1 jcm-09-00715-t001:** Clinical characteristics of the sarcoidosis cohort in total and depending on Anti-NELF-E positivity.

Characteristic	All (*n* = 82)	Anti-NELF-E IgG (+) (*n* = 29)	Anti-NELF-E IgG (−) (*n* = 53)	*p*
Age, years (IQR)	48 (39–59)	42 (37–54)	51 (42–59)	0.123
Female, *n* (%)	35 (43)	13 (45)	22 (42)	0.771
Any pulmonary involvement, *n* (%)	72 (88)	25 (86)	46 (88)	0.767
Extrapulmonary involvement, *n* (%)	48 (59)	18 (62)	30 (58)	0.701
Löfgren-syndrome	18 (22)	7 (24)	11 (21)	0.757
Xray type				0.041
0	10 (12)	4 (14)	6 (12)	
I	7 (9)	0	7 (13)	
II	39 (48)	18 (62)	21 (40)	
III	10 (12)	5 (17)	5 (10)	
IV	15 (19)	2 (7)	13 (25)	
Chronic course	66 (84)	23 (82)	43 (84)	0.803

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
