# Peer review of "Presence of Antibodies Binding to Negative Elongation Factor E in Sarcoidosis"

_jcm, 2020, doi:10.3390/jcm9030715_

Round 1

Reviewer 1 Report

In this manuscript, Baerlecken et al. evaluated the presence of autoantibodies in patients with sarcoidosis. They identified Elongation Factor E (NELF-E) as an autoantigen in a significant fraction of patients. Please find below a list of points.

  • Figure 1

While 10/24 (42%) patients of the initial derivation cohort display detectable levels of antibodies against NELF-E, 3/3 (100%) of patients who had their sera screened were positive for antibodies against NELF-E.

How were these three patients selected for protein array?

  • Figure 2: please use arrows or arrowheads referring to the figure legend to guide the reader.

  • Figure 3

No significance is achieved in the validation cohort regarding IgG-antibodies against the NELF-E comparing patients with sarcoidosis and healthy controls (right panel). I would highly encourage the authors to increase the number of HC (n=7) to reach significance. Otherwise these data do not confirm the authors’ finding. 

Minor points:

  • Line 63 “For the derivation cohort we collected the sera of patients with sarcoidosis (n=25).” Line 119 “We included a total of 82 patients with a definite diagnosis of sarcoidosis (first derivation cohort n=24”. Please confirm the number of patients included.

  • Line 141 “as fare as possible in”. Please correct typo.

  • Line 195 “NELF-E as antigen itself might not be involved as an antigen…”. This sentence is unclear and may need to be improved.

Author Response

In this manuscript, Baerlecken et al. evaluated the presence of autoantibodies in patients with sarcoidosis. They identified Elongation Factor E (NELF-E) as an autoantigen in a significant fraction of patients. Please find below a list of points.

Figure 1

While 10/24 (42%) patients of the initial derivation cohort display detectable levels of antibodies against NELF-E, 3/3 (100%) of patients who had their sera screened were positive for antibodies against NELF-E.

How were these three patients selected for protein array?

Response: All of the selected patients had highly active disease and a Loefgren syndrome.  These patients were considered to most likely yield immunological features especially because of the acute onset. We added this to the method section, subsection 2.2. Protein array of our manuscript (page 2):

Three different sera of patients with acute sarcoidosis presenting with Loefgren’s syndrome at disease onset were screened for the presence of autoantibodies by high-density protein array (hEx1, Source BioScience LifeSciences, Germany) and subsequently compared against existing screens of patients with inflammatory or autoimmune disease. These patients were initially chosen due to their characteristic clinical features and brief disease history.

Figure 2: please use arrows or arrowheads referring to the figure legend to guide the reader.

Response: We thank the reviewer for this suggestion and adapted the figure and figure legend accordingly.

Figure 3: No significance is achieved in the validation cohort regarding IgG-antibodies against the NELF-E comparing patients with sarcoidosis and healthy controls (right panel). I would highly encourage the authors to increase the number of HC (n=7) to reach significance. Otherwise these data do not confirm the authors’ finding.

Response: We thank the reviewer for pointing out this shortcoming of our manuscript. We collected serum samples from another 6 healthy controls from our departments and performed the anti-NELFe ELISA on the samples and added the results to the manuscript. The results of all 13 healthy controls are now displayed in the Figure 3B. After increasing the number of healthy controls, the p-value comparing the titers between the sarcoid cohort and healthy controls via Wilcoxon test was 0.0012 and there were no healthy controls which were above the serum cut-off. The titers were overall lower compared to the other healthy controls. To exclude a relevant “batch effect”, we re-tested a subset of sarcoidosis samples in the second ELISA with no change concerning the cut-off value (i.e. the patients who were positive in the first ELISA were also positive in the second run and vice versa).

Minor points:

Line 63 “For the derivation cohort we collected the sera of patients with sarcoidosis (n=25).” Line 119 “We included a total of 82 patients with a definite diagnosis of sarcoidosis (first derivation cohort n=24”. Please confirm the number of patients included.

Response: There was a typo in Line 63; the correct number of sarcoidosis patients in the derivation cohort was n=24. We corrected this mistake and thank the reviewer pointing it out.

Line 141 “as fare as possible in”. Please correct typo.

Response: Typo was corrected.

Line 195 “NELF-E as antigen itself might not be involved as an antigen…”. This sentence is unclear and may need to be improved.

Response: We agree with the reviewer that the sentence needs revision and adjusted the sentence as follows: “It is noteworthy that NELF-E is not expressed as an extracellular antigen and thus cannot be targeted by the immune system with the tissue being intact. NELF-E is likely released as an antigen after the damage of cells within inflamed tissue.”

Reviewer 2 Report

In this manuscript Baearlecken et al. describe the presence of antibodies against Negative Elongation Factor E (NELF-E) in a fraction of sarcoidosis patients. This novel discovery was made using 3 sera from acute sarcoidosis patients taken before treatment start using a protein array and was absent in previously performed arrays from a variety of other diseases. As confirmation the authors used ELISA testing of two cohorts of a total of 82 sarcoidosis patients and identified 29 as NELF-E antibody positive indicating a possible relevance of NELF-E autoantibodies in sarcoidosis.  As this read out of the ELISA is central to the presented results, I would like to suggest a few doubts and ask for clarifications regarding this technique and the further evaluation of this finding.

Questions ELISA Results:

The IgG titers identified as positive especially in Figure 3B are very close to the cut of serum. In A sarcoidosis patient titer’s look like 2 populations, why is it more of a continuum in B?  Are these really two distinct populations of NELF-E positive/negative in B? Can another independent method show a clearer distinction? I am afraid that this finding can be an artefact especially as in figure 4 all samples move up with increasing lung involvement.

Standard: Why do the authors use the serum from patient as standard, any other possibility? = Is that one of the positive identified 3 serums in the protein array?  Always the same?

Cut off serum: How is the cut of serum-line defined? Is that for healthy control/pooled healthy controls+3(?) SD? As this is essential for the definition of NELF+/- patients in the current method (as not two distinct populations are visible) a good description/evaluation of this parameter is needed.

Questions relevance/discussion:

Are there any other disease known diseases with NELF antibodies?

The authors describe an increase of the proportion of NELF+ sarcoidosis patients with increased  lung involvement, could any other distinctions between NELF+/- patients be found? Maybe regarding the pathology?

The authors speculate in the discussion that NELF antibodies might not be involved in the onset but a bystander phenomenon. Can the authors give any speculations on the kinetics as antibodies are also found in Löfgren’s syndrome? Do the authors see any differences in NELF protein distribution in tissues between NELF+/- sarcoidosis patients?

The authors even give the interesting suggestion that the antibodies might have been generated in response to Mycobacterium absessus infection and cross-reactive to NELF-E. Can this be evaluated further as the development of NELF-E antibodies is not essential to develop sarcoidosis?

Minor comment:

Mistake description results figure 3: 29/82 patients should be 35% not 28% as stated in text (line 144)

Author Response

In this manuscript Baearlecken et al. describe the presence of antibodies against Negative Elongation Factor E (NELF-E) in a fraction of sarcoidosis patients. This novel discovery was made using 3 sera from acute sarcoidosis patients taken before treatment start using a protein array and was absent in previously performed arrays from a variety of other diseases. As confirmation the authors used ELISA testing of two cohorts of a total of 82 sarcoidosis patients and identified 29 as NELF-E antibody positive indicating a possible relevance of NELF-E autoantibodies in sarcoidosis.  As this read out of the ELISA is central to the presented results, I would like to suggest a few doubts and ask for clarifications regarding this technique and the further evaluation of this finding.

Questions ELISA Results:

The IgG titers identified as positive especially in Figure 3B are very close to the cut of serum. In A sarcoidosis patient titer’s look like 2 populations, why is it more of a continuum in B?

Response: The second cohort was measured after the initial derivation cohort and thus these results may be a consequence of the so called “batch effect”. In order to address concerns raised by reviewer#1, we used another ELISA run to measure new healthy controls (total n=13, displayed in Figure 3B)  and re-measured a subset of available sera from the confirmation cohort. It is obvious, that the titers in the new ELISA were somewhat lower than in the first ELISA, but within the new healthy controls, there were also titers within the range of the first ELISA. Within the re-measured sarcoidosis samples, there was no change with regards to the cut-off, while also the titers were overall slightly lower. The distribution of positivity was unchanged however, so we did not include the new values in the manuscript.

Are these really two distinct populations of NELF-E positive/negative in B? Can another independent method show a clearer distinction? I am afraid that this finding can be an artefact especially as in figure 4 all samples move up with increasing lung involvement.

Response: We would like to point out, that the NELF-E ELISA was developed in our laboratories and there is no commercially available alternative test available to date. While developing another method is principally feasible, we are afraid that the time that was given to us to submit a revised version of this manuscript of just 5 days does not allow us to develop that new technique.

Standard: Why do the authors use the serum from patient as standard, any other possibility? = Is that one of the positive identified 3 serums in the protein array?  Always the same?

Response: We agree that using a patient serum as standard is not without limitations, but it is a very commonly done in tests for detecting autoantibodies using ELISA. As an example, in the universally used anti–citrullinated protein antibody ELISAs for rheumatoid arthritis, the cut off is defined as the mean plus 3 standard deviations of healthy controls. The serum (patient or healthy control) close to that point is usually chosen as cut off serum. In our setting, after several tests, the chosen serum of the patient (indeed one of the used sera for the protein array) was the most reliable method.

Cut off serum: How is the cut of serum-line defined? Is that for healthy control/pooled healthy controls+3(?) SD? As this is essential for the definition of NELF+/- patients in the current method (as not two distinct populations are visible) a good description/evaluation of this parameter is needed.

Response: We thank the reviewer for pointing out the cut off needs further definition in the method section. For the definition of a cut off, sera of 16 patients with sarcoidosis, 16 with infectious diseases and 16 blood donors were measured in duplicate. The optimal cut off to differentiate sarcoidosis patients and controls was measured by ROC analysis. The serum of a sarcoidosis patient which provided an OD exactly at that cut off was used as cut off control in all further assays. We added this more detailed description to the method section.

Questions relevance/discussion:

Are there any other disease known diseases with NELF antibodies?

Response: To our knowledge and literature research, presence of NELF antibodies have not been described for any other disease, which we think renders the findings presented herein novel and encouraging for further research. We added this to the first paragraph of the Discussion section of our manuscript: “. To our knowledge, the presence of NELF-E antibodies has not been described in the context of other diseases to date.”.

The authors describe an increase of the proportion of NELF+ sarcoidosis patients with increased lung involvement, could any other distinctions between NELF+/- patients be found? Maybe regarding the pathology?

Response: We thank the reviewer for this important question: We retrospectively evaluated the available data on these patients and could not find significant differences with regards to presence of NELF-E antibodies for age, gender, extrapulmonary involvement and apparent chronicity of the disease suggested by presence of Loefgren’s syndrome. Regarding histology data, it is important to note, that the 2 cohorts were derived from different departments: the first cohort (derivation cohort) was recruited from the rheumatology department and the second (validation cohort) was derived from the respiratory medicine department and this may cause some bias concerning the work-up of these patients. Histology data was available in 70% of patients from the respiratory department, but tissue sampling was often performed in external hospitals (mostly by EBUS transbronchial needle aspiration) and was thus not reviewed by a reference pathologist and written reports available were often not precise with reference to the presence of non-caseating granulomata. The lack of available actual biomaterial thus limits the comparability of the cohorts. We added this as a limitation of our study to the discussion section: “Furthermore, in many cases tissue samples were not available for review by a reference pathologist and data review was limited to written reports so sound differences between the NELF-E antibody positive and negative cohorts in this regard cannot be reliably assessed.”

The authors speculate in the discussion that NELF antibodies might not be involved in the onset but a bystander phenomenon. Can the authors give any speculations on the kinetics as antibodies are also found in Löfgren’s syndrome? Do the authors see any differences in NELF protein distribution in tissues between NELF+/- sarcoidosis patients?

Response: In cancer, autoimmune phenomena can occur a long time before cancer disease manifests. Autoantibodies are already detected prior to disease outbreak and may be used for risk prediction. Of interest, autoantibodies in cancer are directed against proteins which drive cancer evolution such as TP53 (Park et al. Arch Pathol Lab Med 2011). On this background we speculate that NELF-E may be part of a central pathway involved in sarcoidosis evolution. Of course, autoantibodies may be only a bystander phenomenon without any functional role; further research is needed to establish the precise role of NELF-E antibodies. We acknowledged this in the last paragraph of the discussion: “Further research is required to establish the precise functional role of NELF-E antibodies in sarcoidosis” and previously discussed: “In the cancer field, however, it is well recognized that autoantibodies against driver mutations/ oncogenes precede the manifestation of the cancer and maybe used as a predictive biomarker [29].”. We could not find any differences in the protein distribution given the lack of available tissue material as described in the previous comment.

The authors even give the interesting suggestion that the antibodies might have been generated in response to Mycobacterium absessus infection and cross-reactive to NELF-E. Can this be evaluated further as the development of NELF-E antibodies is not essential to develop sarcoidosis?

Response: In a project in progress, using a peptide with similarity between human NELF-E and a peptide of Mycobacterium abscessus, 6 of 8 shared positivity and 14 of 16 shared negativity, thus there is a hint towards cross reactivity between these two peptides derived from NELF-E and M. abscessus. These results however are preliminary and we do not feel confident enough at this stage to share these results in the manuscript. Again, as the reviewer may agree, the 5-day deadline for this revision does not allow for substantial progress on this data.

Mistake description results figure 3: 29/82 patients should be 35% not 28% as stated in text (line 144)

Response: We thank the reviewer for point out this mistake and corrected it accordingly.

Reviewer 3 Report

Re: JCM-731672

Presence of Antibodies binding to Negative 3 Elongation Factor E in Sarcoidosis

Baerlecken et al studied cohorts of patients with sarcoidosis and healthy individuals and investigated the presence of autoantibodies that could characterize the disease. Their findings identified Negative Elongation Factor E (NELF-E) as a novel autoantigen for sarcoidosis.  

The study is well designed and executed. The manuscript is very well written and comprehensive. However, there are limitations in this study (and the authors acknowledge this in the discussion), which need to be addressed. NELF-3 autoantibodies were detected in only a third of sarcoidosis patients, a percentage not sufficient to define them as characteristic of the disease. There are also no functional data linking NELF-3 autoantibodies to sarcoidosis that would strengthen the authors’ main conclusion. Overall, the findings of the study, although well presented, do not convince that sarcoidosis could be an autoimmune disease or that NELF-3 autoantibodies could be used to diagnose sarcoidosis.

Author Response

Baerlecken et al studied cohorts of patients with sarcoidosis and healthy individuals and investigated the presence of autoantibodies that could characterize the disease. Their findings identified Negative Elongation Factor E (NELF-E) as a novel autoantigen for sarcoidosis.

The study is well designed and executed. The manuscript is very well written and comprehensive. However, there are limitations in this study (and the authors acknowledge this in the discussion), which need to be addressed. NELF-3 autoantibodies were detected in only a third of sarcoidosis patients, a percentage not sufficient to define them as characteristic of the disease. There are also no functional data linking NELF-3 autoantibodies to sarcoidosis that would strengthen the authors’ main conclusion. Overall, the findings of the study, although well presented, do not convince that sarcoidosis could be an autoimmune disease or that NELF-3 autoantibodies could be used to diagnose sarcoidosis.

Response: We thank the reviewer for his appraisal of our manuscript and fully agree with his notion that our data (at this stage) is insufficient to prove an autoimmune pathogenetic role of NELF-E antibodies in sarcoidosis. We rephrased our discussion to acknowledge the hypothesis-generating character of our results: “We fully acknowledge that our findings at this stage are primarily hypothesis generating and do not establish a clear link between autoimmunity and sarcoidosis and given their presence in only a subset of sarcoidosis patients, cannot function as a diagnostic tool at this stage.”. We also adapted the final paragraph of our introduction, previously reading: “This is the first study showing clear evidence of autoimmune phenomena in sarcoidosis” to “… demonstrating the presence of NELF-E antibodies in a specific disease”, since their presence has not been described to our best knowledge in any other disease to date. The exact pathogenetic role of NELF-E antibodies in sarcoidosis are yet to establish and lie beyond the scope of this hypothesis generating manuscript. We thank the reviewer for highlighting the limitations of the data and hope our modifications to the conclusions are now in keeping with the results.

Round 2

Reviewer 2 Report

I thank the authors for their extensive and valuable answers. I completely understand that within 5 days no major  experiments can be performed and have no further comments. 

Reviewer 3 Report

The authors made sufficient changes in their discussion and introduction sections and the revised version of the manuscript is generally improved.

Please check line 216.